# Compulsive and compensative buying among online shoppers: An empirical study

**Grzegorz Adamczyk**◉*

Department of Sociology, Catholic University of Lublin, Lublin, Poland

* grzegorz.adamczyk@kul.pl

## Abstract

The present study examines the phenomenon of compensative and compulsive buying among online shoppers. Firstly, the obtained empirical data make it possible to estimate the prevalence of compensative and compulsive buying among the general population of Poles aged 15 years old and over, with the sample split into users and non-users of the e-commerce market offer. Secondly, the conducted analysis shows to what extent the prevalence of compulsive and compensative buying is differentiated by the frequency of online shopping, by the extent of the expenditures on online shopping compared with offline shopping, by attitudes towards online shopping, and by sociodemographic conditions (gender, age, monthly net income of household). The findings come from a survey conducted in 2019 based on a nationwide statistically representative sample of 1,000 Poles aged 15 years old and over. Drawing on this survey based on the German Compulsive Buying Indicator (GCBI), the prevalence of compulsive buying is observed at about 3% and compensative buying at about 12%. Dividing the general population into online and offline shoppers, one can see serious differences between both target groups; the share of compulsive and compensative buyers in the segment of online shoppers amounts to 3.6% and 16.9%, while among non-online shoppers– 3.3% and 10.1%. The strongest susceptibility to compulsive buying is characteristic of female online shoppers having very positive attitudes towards online shopping and doing online shopping very frequently.

## Introduction

For recent years researchers of consumer behaviours have observed significant changes of buying habits resulting in a stronger tendency to online shopping in terms of its frequency and value. According to the European Commission, the share of e-sales turnover in the total turnover of enterprises in the EU increased from 14% to 18% between 2010 and 2018. The dynamic was differentiated depending on the country. E.g. in Bulgaria, the increase of the coefficient was noticed in a modest range from 2% to 4%, whereas in Belgium from 18% to 33%. In Poland, the growth of the coefficient from 8% to 18% was greater than the EU's average [1]. The change of shopping habits lies behind the tendencies. In 2019 almost 2/3 of the EU consumers declared at least one purchase online in the last 12 months (63%), which means an increase by 23 percentage points compared with 2010. The greatest growth of the shopping

**Data Availability Statement:** The data are available from the public repository of the Catholic University of Lublin (http://hdl.handle.net/20.500.12153/1676) A commercial usage of the data is excluded.

**Funding:** This work was supported by the Institute of the Public Opinion and Market Research GfK Polonia Sp. z o.o. belonging to the GfK Group. The funders had no role in study design, analysis, decision to publish, or preparation of the manuscript. The role of the funder consisted in the data collection (1,000 of direct interviews on the statistically representative sample of Poles aged 15 years and over).

**Competing interests:** The author has declared that no competing interests exist. This does not alter our adherence to PLOS ONE policies on sharing data and materials.

buyers' segment was evidenced in Eastland (from 17% to 68%). The modest increase from 71% to 82% was observed in Norway, where the majority of consumers use the e-commerce market offer either way. In Poland, a considerable growth of the online shoppers' ratio from 29% to 54% was pointed out [2]. Certainly, the COVID-19 pandemic has contributed to a further dynamic development of the e-commerce market offering safer shopping than the stationary retail.

The growth of the e-commerce market might be accompanied by a simultaneous development of some disorders connected with buying. This statement concerns compensative and compulsive buying first of all which can accompany online shopping to a greater extent than offline shopping. In addition, this kind of shopping behaviour might be a remedy for the negative consequences of the COVID-19 pandemic such as social isolation, experience of threat, unemployment, worsening self-esteem connected with declining intensity of social contacts, limited possibilities of self-actualization etc. These premises base on results of the empirical studies which evidenced sufficiently that shopping online might be an important universal factor of compulsive buying independent of cultural conditions [3–10].

Although disorders connected with buying were noticed at the beginning of 20th century [11, 12], a broader discussion about the nature of the phenomenon began with the study of Faber, O'Guinn, and Krych [13]. Originally, compulsive buying won interest of clinical psychologists and psychiatrists describing compulsive buying as a behavioural addiction or as a disorder of impulse control [14]. Later, consumer researchers entered the circle interested in compulsive buying treating it as an irrational way of purchasing. To search for social factors strengthening compulsive buying in different societies and to seek universal correlations between compulsive buying and social variables characterising the global consumer society belong to the most important tasks of this approach.

Consumer research is a reference point of the study which attempts to answer the following questions:

1. Does the frequency of shopping online and the extent of expenditures on online shopping differentiate susceptibility to compensative/ compulsive buying?

2. Do the attitudes towards shopping online differentiate susceptibility to compensative/ compulsive buying?

3. To what extent do the frequency of shopping online, the attitudes towards shopping online, and the basic sociodemographic variables such as age, gender, household income explain susceptibility to compensative/ compulsive buying?

## Conceptual framework

O'Guinn and Faber define compulsive buying as "chronic, repetitive purchasing that occurs as a response to negative events or feelings" [15, p. 149]. The purchase act of this kind is conducted by compulsive buyers to compensate for the problems in other life areas. Although the consequences of this behaviour are negative psychologically (guilty feelings, weak self-esteem), socially (disordered interpersonal relationships, conflicts), and economically (debts), compulsive buyers experience compulsive buying positively [15].

Mueller, Mitchel, and de Zwaan [14] treat compulsive buying as an excessive focus on shopping turning into a desire to purchase something which is difficult to restrain. With time, compulsive buyers increasingly limit their activities to the purchasing process. The actual use of possessed goods recedes into the background; once purchased, they remain even hidden and unpacked at home because it is only the purchasing act that improves one's own value or compensates for the experienced negative mood. The utility value of purchased consumer goods

plays a secondary role. Finally, the feeling of pleasure and relief provided initially by the purchase act wither and are replaced by the feeling of guilt and remorse when the consequences of the act are perceived. The real subject of compulsive buying is compensation provided by the purchasing act, not by a physical good. Compulsive buying may compensate for stress generated by profession, work or disappointments in everyday life [16].

Certainly, an experience of compensative buying or consumption is nothing unusual in consumer society. Consumption aiming at an improvement of an individual's mood, buying as a consolation or relaxing shopping are common strategies coping with more or less difficult situations of everyday life. Hence, the borderline between compulsive and compensative buying might be seemingly effaced. To explain the differences between both types of consumer behaviour clearly, the Rational Choice Theory seems to be especially suitable. According to Becker [17], consumers choose this behaviour from a set of the accessible options, which maximizes profits and minimizes the incurred costs at the same time. From this point of view, compulsive buying is irrational because costs of the behaviour exceed profits, sometimes to a considerable extent. On the one hand, compulsive buyers experience a relief of tension and an improved self-esteem (profits), while on the other hand the costs are significantly more complex psychologically (loss of self-control, turning back remorse and feeling of guilt, withdrawal symptoms), socially (crisis of interpersonal relationships, interrupted professional career, problems with law), economically (excessive expenditures and household's debts), or even in the area of health (comorbidity of behavioural or substance addiction) [18]. From this point of view, compensative buying is harmless because the negative consequences of the behaviour never exceed the profits such as improved mood. Hence, compensative buying can be treated as an autonomous style of consumer behaviour which may precede compulsive buying [19]. This idea is reflected in the Compulsive Buying Scale (CBS) highlighting three segments of shoppers placed on the continuum: ordinary, compensative, and compulsive buyers [15].

Various psychological and social factors strengthen susceptibility to compulsive buying. For example, self-esteem and materialism are mentioned as important variables in this area. A significantly lower self-esteem experienced by compulsive compared with non-compulsive buyers was evidenced already by O'Guinn and Faber [15]. The subsequent studies supported O'Guinn and Faber's conclusions [20–23]. Roots of an individual's weakened self-esteem might be sought in specific socialisation conditions, authoritarian or overprotective upbringing styles leading to distortion of an individual's autonomy and a healthy feeling of one's own value, as well as in the use of material goods as the upbringing means [24, 25].

A positive correlation between compulsive buying and materialism were evidenced by O'Guinn and Faber [15] and some of their followers [23, 26–29]. Although materialism understood as an overall life orientation is not a new phenomenon, the prevalence of the orientation is especially supported by the development of consumer society. This type of society uses symbolic values of brands [30] to foster the consumerism that fuels the economic growth. It causes different negative side effects. Consumeristically oriented people attaching more importance to the possession of material goods and consumer symbols are less happy and more insecure [31]. Although a steady recurrence of a search for a compensation in the world of consumer goods and shopping is one of the common solutions offered by consumer society for its members, the abuse of the compensation or demonstrative functions of consumer goods might lead to the development of compulsive buying.

## Prevalence

The first European empirical studies referring to compulsive buying were conducted in 1989 in Germany among persons declaring compulsive buying (direct interviews) and inhabitants of Stuttgart (postal questionnaire). The German Compulsive Buying Indicator was used as the

measurement instrument [21]. A nationwide study was conducted in West Germany two years later based on a sample of people aged 14 years and older. The results proved the share of compulsive buyers in the general population on the level of 4%. The study replicated in 2001 pointed out the coefficient on the level of 8% [32].

Using the GCBI, the online study conducted in 2010 in Denmark evidenced the 6% share of compulsive buyers and the 10% share of compensative buyers in the population of 15-84-year-olds [33]. In Austria, where direct interviews were conducted, the size of the compulsive and compensative buyers' segments was observed on the levels of 8% and 19% [34].

A telephone survey based on the Compulsive Buying Scale which was conducted in 2004 in the USA displayed the prevalence of compulsive buying on the level of 6% [35]. Using the GCBI scale, a postal survey conducted in the years 2012–2013 among the 15-65-year-olds from Galicia in Spain showed a clear tendency to compulsive buying in 7% of respondents [27].

An online survey conducted in Eastland displayed the 8% share of compulsive buyers on the basis of the CBS [28]. The study carried out in Hungary using the Edwards Compulsive Buying Scale pointed out the share of compulsive buyers on the level of 2% in the general population aged 18–64 [22]. The further results of the same project showed the ratio of compulsive buyers among customers of some Hungarian shopping centres near 9% [36].

The validity of the previously formulated thesis about cultural independence of the phenomenon named as compulsive buying is confirmed by the numerous empirical studies conducted in Asian countries. Guo and Cai [7] evidenced the prevalence of compulsive buying among 19% of Chinese students aged 12–18 years old coming from middle schools located in four cities: Shanghai, Tianjin, Chongqing, Xiamen and among 25% of Bangkok's residents belonging to the same age and status groups. Similarly high prevalence of susceptibility to compulsive buying among youth and young adults was pointed out among Korean students. The results of the study carried out in Seoul, Busan, Daejeon, and Kwangju showed the 16% share of compulsive buyers among 15–25 year-olds [8]. Lo and Harvey [9] observed susceptibility to compulsive buying among 16% of Taiwanese participants taking part in their experiment about relations between compulsive buying and obsessive acquisition as well as hoarding behaviour. He, Kukar-Kinney, and Ridgway [10] pointed out susceptibility to compulsive buying among 29.1% of respondents recruited among the participants of the Sojump online consumer panel rather representing the urban and younger part of the Chinese population.

## Research aim

In the context of the research questions the following hypotheses for the study are assumed:

H1: Persons doing the shopping online show susceptibility to compulsive buying to a greater extent than persons doing their shopping offline

H2: Susceptibility to compulsive buying grows along with the increasing frequency of shopping online

H3: Persons spending more money on online than stationary shopping show susceptibility to compulsive buying to a greater extent than persons spending more money on stationary than online shopping

H4: The more positive attitudes towards shopping online, the stronger susceptibility to compulsive buying

In general, a correlation between shopping online and compulsive buying is assumed within the study for several reasons. Firstly, the subject of compulsive buying is not a concrete consumer good, but the compensation rooted in a purchase act of the good. Realisation of the

purchase act in stationary retail involves more effort, a shopper has to cover a geographical distance, to find sufficient free time, to spend additional money on transport means etc., while the same purchase act online is significantly easier to conduct because only a computer or smartphone, the Internet connection with a browser, and a service of online banking are required to do the shopping. All the things are commonly used in economically developed societies. As a consequence, purchase acts can be carried out during all parts of a day without a necessity to leave the dwelling place. An easy access to the world of consumer goods renders an easy access to the purchase acts which might deliver the experience of compensation as a response to problems of everyday life.

Secondly, purchase acts carried out online are more anonymous than the purchase acts in stationary retail. Greater anonymity can be encouraging for a buyer to repeat the purchase acts many a time. The buyer in stationary retail has to handle the physical presence of a seller. Even though a buyer visits a self-service store, where anonymity is significantly greater than in traditional retail, very frequent calls on the branch might cause uneasiness on the part of the buyer connected with the possibility of being perceived by the staff or acquaintances met by chance as a person having strange habits. From the psychical point of view, compulsive buying is one of behavioural addictions. Generally, addicted persons, especially on the initial stages of the addiction, try to hide it. On the one hand, they want to protect their social status, e.g. the professional position which could be threatened if the colleagues or people of importance find out the disorder. On the other hand, a lack of awareness on the part of the social environment about the addiction is desirable for the addicted person, they could continue their addiction behaviour without a risk that somebody would like to stop the behaviour. From this point of view, compulsive buying online protects the addicted person against the negative consequences of their addiction being disclosed.

Thirdly, compulsive buying online might help better than shopping in stationary retail to hide the addiction from the closest persons inhabiting the same household. In a way, compulsive buying online protects good personal relationships longer than compulsive buying carried out in stationary retail. Conducting the measurements connected with compulsive buying such as checking the offer, ordering, payment, picking up the goods is easier in online retail than in stationary retail. The addicted person staying at home with relatives can use the browser, send the electronic ordering form, pay bills and order the goods with delivery to a parcel locker. In turn, the addiction might remain unnoticed by home dwellers longer and it can develop faster.

The results of different previously conducted studies confirm the above presented assumptions. Trotzke at al. [5] using the Compulsive Buying Scale (CBS) and the Short Internet Addiction Test Modified for Shopping (s-IAT shopping) on the sample of 240 female participants aged 18–64 years old found out clear correlations between online pathological buying and addictive Internet usage. Firstly, both measurement instruments turned out as relatively strongly correlated (r = 0.517, p < 0.001); secondly, an overlap of compulsive buying and addictive usage of the Internet for shopping was observed–almost 1/5 of 47 participants susceptible to compulsive buying or addicted to the Internet showed both features at the same time.

Wang and Yang [4] who analysed the results of a survey conducted among 403 mostly undergraduate students from a Taiwanese university confirmed two hypotheses about a positive correlation between compulsive buying and passion about online shopping understood as a strong inclination by individuals to take part in the activity, as well as between compulsive buying and dependency on shopping online. An increasing obsessive passion about online shopping clearly goes along with the growing susceptibility to compulsive buying (r = 0.34; p < 0.001). The same direction appears in case of the second correlation–an increasing addiction to online shopping co-occurs with the growing susceptibility to compulsive buying (r = 0.20; p < 0.001). In addition, the results of the t-test for independent samples show

without doubts that compulsive buyers are statistically significantly more strongly addicted to online shopping activities than non-compulsive buyers (t(404) = -3.47, p < 0.001).

Bighiu, Manolica, and Roman [6] conducted a survey on the sample of 100 mostly undergraduate students of Al Ioan Cuza University in Romania. The data obtained on the basis of the Compulsive Buying Scale show the share of compulsive buyers in the sample on the level of 13%. Although all respondents do the shopping online, the students qualified as compulsive buyers carry out a little bit more purchase acts annually on average than non-compulsive students.

200 students took part in the survey conducted by Duroy, Gorse, and Lejoyeux [37], who measured the prevalence of compulsive buying, motivations to shop in the Internet, and correlations of compulsive buying with other addictions among Parisian young adults. Using the Echeburùa's clinical screener, the prevalence of online compulsive buying was measured on the level of 16%. Clearly, online compulsive buyers do the shopping online statistically significantly more frequently than other students (private sales websites: t = 1.57, p < 0.001; mobile phone: t = 2.98, p < 0.001). In addition, it turned out that online expenses borne by compulsive buyers were statistically significantly higher than the expenses of other students. The same conclusion was drawn for the time extent used to do the shopping online. While compulsive buyers spent 51 minutes on online shopping daily and 30 minutes nightly on average, the non-compulsive buyers spared only 13/ 5 minutes on average on the activities.

Raab and Neuner [3], who conducted a study on the sample of 423 Ebay users aged 18–40 years old, verified the hypothesis assuming a positive correlation between the Internet addiction and compulsive buying. The results of the study evidenced without doubts that a statistically significant correlation between both variables exists (r = 0.45; p < 0.001). In addition, compulsive buyers did the shopping online via the Ebay service more frequently and spent more money on the shopping than the users of the Ebay service who were not classified as compulsive buyers.

In the light of the above presented data and observations, the assumption about the stronger susceptibility to compulsive buying of online shoppers than offline shoppers seems to be justified.

## Materials and methods

The survey about compulsive buying among users of the e-commerce offer was conducted in April 2019. Using the method of the Computer Assisted Personal Interview at the respondent's home, the study was carried out on the statistically representative sample of 1,000 Poles aged 15 years old and over (Table 1). Once the survey had been approved by the Ethical Committee of the Institute of Sociological Sciences at the Catholic University of Lublin (decision 1/2019), the fieldwork was conducted by the Institute of Market and Public Opinion Research GfK Polonia which was responsible for the recruitment process of participants including their verbal consents to take part in the survey. The verbal consents of participants were ticked in the questionnaire by interviewers.

The following table presents the sociodemographic structure of the sample.

The tendency to compulsive buying was measured by the German Compulsive Buying Indicator [21], which is a modified version of the Canadian Compulsive Buying Measurement Scale proposed by Valence, d'Astous and Fortier as one of the first empirical tools devoted to compulsive buying. In the beginning, it consisted of 16 items describing four basic dimensions of compulsive buying–tendency to spend, reactive aspect, postpurchasing guilt, and family environment. Finally, the latter one was excluded because of a weak internal cohesion and 13-items scale found acceptance [38]. Schernhorn, Reisch, and Raab [21] proposed the

**Table 1. Sociodemographic characteristics of the sample.**

|  | *N =* | % |
|---|---|---|
| **Gender** | | |
| Male | *479* | 47.8 |
| Female | *521* | 52.1 |
| **Age** | | |
| 15–29 | *255* | 25.5 |
| 30–39 | *108* | 10.8 |
| 40–49 | *199* | 19.9 |
| 50–59 | *179* | 17.9 |
| 60 + | *260* | 26.0 |
| **Household size** | | |
| Only respondent | *116* | 11.6 |
| 2 persons | *220* | 22.0 |
| 3 persons | *232* | 23.2 |
| 4 persons | *214* | 21.4 |
| 5 persons and over | *218* | 21.8 |
| **Children in household** | | |
| Yes | *312* | 31.2 |
| No | *688* | 68.8 |
| **Education** | | |
| Primary | *240* | 24.0 |
| Vocational | *234* | 23.4 |
| Secondary | *341* | 34.1 |
| Higher | *184* | 18.4 |
| **Professional status** | | |
| Managers/ Directors/ Entrepreneur | *61* | 6.1 |
| White collar | *189* | 18.9 |
| Blue collar | *265* | 26.5 |
| Farmers | *105* | 10.5 |
| Students | *90* | 9.0 |
| Pension | *207* | 20.7 |
| Housewife | *53* | 5.3 |
| Other | *30* | 3.0 |
| **Monthly net income of household (USD 1 = approx. PLN 3.75)** | | |
| Up to PLN 2,999 | *174* | 17.4 |
| PLN 3,000–4,499 | *282* | 28.2 |
| PLN 4,500–5,999 | *313* | 31.3 |
| PLN 6,000 and over | *231* | 23.1 |
| **Size of locality** | | |
| Village | *396* | 39.6 |
| Town up to 50,000 inhabitants | *244* | 24.4 |
| Town 50.000–500.000 inhabitants | *251* | 25.1 |
| Town above 500.000 inhabitants | *110* | 11.0 |

Source: Researcher's own study

German version of the scale. This tool is composed of 16 items assessed by respondents on the four-points scale from 1 ("I don't agree") to 4 ("I totally agree"). Faber and O'Guinn [19] proponed the Compulsive Buying Scale comprising seven items referring to typical symptoms of

compulsive buying: loss of impulse control, concealing problems, raising tension when shopping is not possible, reduction of tension based on purchase acts. While O'Guinn and Faber's Compulsive Buying Scale [15] is focused on the measurement of the frequency of displayed behaviours or feelings (6 from 7 items), the Canadian Compulsive Buying Measurement Scale and the German Compulsive Buying Indicator prove susceptibility to compulsive buying to a greater extent.

Before the scale of GCBI was used to measure the prevalence of compensative and compulsive buying in Poland, the one-dimensionality, reliability and normal distribution of the results on the scale had been checked. Firstly, the GCBI scale appeared as one-dimensional (KMO above 0.5; Significance of Barlett's Test of Sphericity below 0.05). Secondly, the GCBI scale showed a high degree of internal consistency achieving a very satisfying degree of reliability at the same time (Cronbach's Alpha = 0.938). Thirdly, the coefficients of skewness and kurtosis indicated that the distribution of the GCBI scale is approximately normal because neither of the values exceeds the interval between -1 and +1 (Skewness = 0.267; Kurtosis = -0.325).

As indicated in hypothesis 4, a co-existence of susceptibility to compensative/ compulsive buying and positive attitudes to online shopping is expected. To measure the last one the question including eight statements describing the attitudes towards shopping online was used (1. I do the shopping online to make a good impression on friends thanks to a cool purchase; 2. Shopping online is very fashionable now; 3. Shopping online is for me the most convenient way to buy different services and products; 4. The Internet serves me as a source of information first of all and I do the shopping in stationary stores in most cases; 5. I watch products firstly in a stationary store very often and then I buy them online; 6. I am not an enthusiast of online shopping, I prefer traditional shopping; 7. I like doing the shopping neither online nor in a traditional way; 8. I prefer shopping in stationary stores–you can go outside of home at least and take some exercise). Each of the statements was measured on the same 5-points Likert's scale (1- I do not agree at all, 2- Rather I do not agree, 3- Neither I agree, nor I do not agree, 4- Rather I agree, 5- I agree completely).

On the first step, the statements were exposed to the factor analysis that showed three components including the following items: component 1 –statements 1 (0.827), 3 (0.761), and 5 (0.859); component 2 –statements 6 (0.672), 7 (0.847), and 8 (0.540); component 3 –statements 2 (0.721) and 4 (0.722). Component 1 is characterised by the highest range of explained variances (35%) and the highest reliability measured by the Cronbach's Alfa = 0.828. Hence, the statements served the segmentation of respondents concerning their attitudes towards shopping online on the second stage.

Each respondent was assigned to one of four segments based on the following rule. If a respondent agreed with 1 or 2 or 3 statements more or less but they did not agree with more than 1 statement at the same time, they were defined as segment 1 of the "fans of online shopping". Who agreed only with one statement and did not agree with two statements or was undecided concerning all three statements belongs to the segment of the "potential fans". Who did not agree with one or two statements and was undecided concerning one or two statements represents the segment of "sceptics". Finally, if a respondent did not agree with any statement, they belong to the segment of "critics".

The following analytical methods will be used on the stage of the statistical analysis of the obtained data. Co-exist of compensative/ compulsive buying and different dimensions of online shopping (frequency, expenditures, overall attitudes) will be analysed based on the t-student test for two independent samples (in case of dichotomous variables) and the one-way ANOVA analysis (in case of ordinal variables). The function of frequency of online shopping, attitudes towards online shopping, age, gender, and income in the prediction of compulsive buying will be examined based on the linear stepwise regression analysis.

## Results

### Prevalence of compulsive and compensative buying

Following Faber and O'Guinn [15, 19], those respondents should be qualified as compulsive buyers who achieve the results on the CBS scale at least equal to two standard deviations above the mean. The respondents who achieve the result between one and two standard deviations above the mean should be defined as compensative buyers. In the presented study, the mean value on the GCBI scale achieved 31.3877 and the standard deviation 9.3482. For this reason, those respondents were qualified as compulsive buyers who achieved the result at least on the level of 50; 3.4% of such cases in the general population were found (3.6% among online shoppers and 3.3% among non-online shoppers). Then, those respondents who achieved the result of 41–49 points were qualified as compensative buyers; 12.4% of these cases in the whole sample were pointed out (16.9% among online shoppers and 10.1% among non-online shoppers). In general, 15.8% of Poles aged 15 years old and over display a tendency to compensative or compulsive buying (20.5% among online shoppers and 13.4% among non-online shoppers).

### Co-occurrence of shopping online and compensative/ compulsive buying

**Shopping online.** Already the first steps of the data analysis show that susceptibility to compulsive buying is differentiated by the fact if consumers prefer online or offline shopping. Online shoppers (those who bought something online at least once in the last year) achieved the average value of 33.44 on the GCBI scale in comparison with the value of 30.38 in case of offline shoppers (those who bought nothing online in the last year). The t-student test for two independent samples makes it possible to reject hypothesis zero that the mean values on the GCBI scale for online and offline shoppers are equal (t(998) = 4.925, p < 0.001). Hence, hypothesis H1 is confirmed—online shoppers show a stronger susceptibility to compulsive buying than offline shoppers.

To prove the correlation between frequency of online shopping and susceptibility to compulsive buying, the respondents were divided into three segments: heavy users of shopping online, medium users, and light users of shopping online. The heavy users buy online at least one product category at least once a month, while the medium users purchase online at least one product category a few times a year. The light users buy online at least one product category only 1–2 times a year or more seldom. Almost 1/10 of the general population belong to the heavy users' segment (9.2%), somewhat more than 1/10 of the population reinforce the medium users' segment (13.6%), and less than 1/10 of Poles represent the light users (8.2%). The non-users' segment embraces more than 2/3 of the general population (68.9%).

The obtained results show a clear tendency that a growing susceptibility to compulsive buying goes along with an increasing frequency of shopping online. The mean value on the GCBI scale increases in the particular segments which describe the growing frequency of shopping online (non-users: 30.26; light users: 32.38; medium users: 34.01; heavy users: 35.04). The results of the crosstab analysis limited to the online shoppers confirm the tendency (Table 2). The shares of compulsive and compensative buyers increase along with the growing frequency of online shopping.

The highest percentage of compulsive buyers was found out among the heavy users of online shopping, whereas the lowest percentage in the segment of light users. The rule is the same in case of compensative buyers. The highest percentage of them was observed among the heavy users, while the lowest ratio among the light users of online shopping.

Using the one-way ANOVA analysis, the statistical significance of differences of mean values between the particular frequency groups on the GCBI scale were proved. Before the

**Table 2. Compulsive and compensative buyers by frequency segments.**

| Segment | (n) | Compulsive buyers | Compensative buyers | Other buying styles | Total |
|---|---|---|---|---|---|
| General population | (1,000) | 3.3% | 12.5% | 84.2% | 100.0% |
| General population of online shoppers | (331) | 3.3% | 17.2% | 79.5% | 100.0% |
| Light users | (102) | 0.0% | 6.9% | 93.1% | 100.0% |
| Medium users | (136) | 4.4% | 18.4% | 77.2% | 100.0% |
| Heavy users | (93) | 5.4% | 26.9% | 67.7% | 100.0% |

Source: Researcher's own study

analysis starts the number of cases within the particular frequency groups were equalled based on a random selection of the cases. The mean values on the GCBI scale for each randomly formed frequency group do not differ significantly from the mean values for the whole frequency groups: non-users– 30.09; light users– 32.62; medium users– 33.58; heavy users– 34.95. Because the significance of the Levene's test equalling $p = .001$ evidences dissatisfying statistical significance of the difference between the variance in the tested groups, the Welch's correction was introduced. The result of the one-way ANOVA indicates at least one pair of mean values differing from each other statistically significantly: $F(3,315) = 4.268$, $p = 0.006$. Hence, $H_0$ assuming that mean values on the GCBI scale do not differ in the examined frequency groups is rejected. The post hoc test shows that the mean values achieved by non-users and heavy users on the GCBI scale differ from each other in a statistically significant way (Mean difference = -4.86, $p < 0.001$). Hence, the hypothesis H2 assuming that susceptibility to compulsive buying grows along with the increasing frequency of shopping online is confirmed in part. Susceptibility to compulsive buying primarily differs between non-users and heavy users of shopping online.

**Expenditure on shopping online.** Online shoppers still spend more money on shopping offline than on shopping online, although the proportion between online and offline expenditures varies among the population of online shoppers. In total, 6% of online shoppers declare they spend online decidedly more than offline, the further 8.2% of online shoppers confirm they spend a little bit more online than offline. Nearly 1/4 of online shoppers perceive a balance between expenditures in both shopping channels (22.1%). Almost 2/3 of online shoppers spend more offline than online, more than a half of them spend decidedly less online than offline. The data indicate a growing susceptibility to compulsive buying along with an increasing level of expenditure online, which is evidenced by the results of the crosstab analysis (Table 3). The share of compulsive and compensative buyers increases along with the growing extent of online expenditures in comparison with offline spending.

The highest percentage of compulsive buyers was found out among those who spend more money on shopping online than on shopping offline. On the contrary, the lowest percentage of

**Table 3. Compulsive and compensative buyers by spending groups.**

| Spending groups | (n) | Compulsive buyers | Compensative buyers | Other buying styles | Total |
|---|---|---|---|---|---|
| General population | (1,000) | 3.3% | 12.5% | 84.2% | 100.0% |
| General population of online shoppers | (331) | 3.3% | 17.2% | 79.5% | 100.0% |
| Expenditures offline exceed expenditures online | (211) | 1.9% | 10.0% | 88.1% | 100.0% |
| Equality between online and offline expenditures | (74) | 4.1% | 25.7% | 70.2% | 100.0% |
| Expenditures online exceed expenditures offline | (46) | 10.9% | 37.0% | 52.2% | 100.0% |

Source: Researcher's own study

compulsive buyers was pointed out among more traditional shoppers whose expenditures in the traditional shopping channels predominate. The same tendency is observable in case of compensative buying. Only 1/10 of shoppers who spend more offline than online show susceptibility to compensative buying, while the ratio of compensative buyers among shoppers allocating more money for online than offline shopping is more than three times higher. In total, nearly a half of Poles spending more money online than offline buy compulsively or compensatively, whereas the rate declines along with the change of the proportion between online and offline shopping.

The analysis of mean values on the GCBI scale confirms the above presented findings: they clearly grow in the increasingly spending groups (expenditures offline exceed expenditures online: 31.20; expenditures offline equal expenditures online: 36.81; expenditures offline exceed expenditure online: 38.29). Again, the one-way ANOVA analysis was used to prove if the differences of mean values between the spending groups on the GCBI scale are statistically significant. Similarly to the previously introduced procedure, the number of cases within the particular spending groups were equalled initially based on a random selection of the cases. The mean values of the each randomly formed spending group do not differ significantly from the mean values of the whole spending groups: expenditures offline exceed expenditures online: 31.78; expenditures offline equal expenditures online: 36.62; expenditures offline exceed expenditures online: 38.29. The significance equalling $p = 0.249$ of the Levene's test makes it possible to acknowledge hypothesis $H_0$ assuming that the difference of the variance in the tested groups is not statistically significant. This result of the one-way ANOVA points out to at least one pair of mean values differing from one another statistically significantly: $F(2,133) = 7.688$, $p < 0.001$. Hence, $H_0$ assuming that the mean values on the GCBI scale do not differ in the examined spending groups is rejected.

The post hoc test shows that some of the mean value pairs differ from one another in a statistically significant way. Those who spend more online than offline achieve a statistically significantly higher mean value on the GCBI scale than those who spend more offline than online (Mean difference = 6.51, $p < 0.001$). The same conclusion can be drawn comparing the mean values of the group spending online and offline the same amount of money with the mean values of those who spend more offline than online (Mean difference = 4.84, $p < 0.001$). It means that persons spending online more than offline and persons spending similar amounts of money online and offline show a stronger susceptibility to compulsive buying than persons whose online expenditures are lower than expenditures in the stationary retail.

**Attitudes towards shopping online.** The variable describing attitudes towards shopping online differentiates results on the GCBI scale to a noteworthy extent.

According to the expectations, those achieve the highest mean value on the GCBI scale who are familiar with online shopping to the greatest extent. Persons unwilling to do this form of shopping are on the opposite pole. The average values on the GCBI scale clearly increase along with more positive attitudes towards online shopping (critics: 28.81; sceptics: 31.50; potential fans: 32.19; fans of online shopping: 37.00). Results of the data analysis in the cross tabs (Table 4) confirm without doubts that the growing positive attitudes towards online shopping go with a stronger susceptibility to compulsive/ compensative buying.

The above presented data indicate that fans of online shopping display the strongest susceptibility to compulsive buying–about 1/20 of the segment achieve the results on the GCBI scale that allow to suppose susceptibility of the consumers to compulsive buying. Critics and potential fans are on the opposite pole–about 2% of them show susceptibility to compulsive buying. The tendency is clearer concerning compensative buying. The share of this type of consumers grows along with the segments characterised by more positive attitudes towards online shopping. Whereas less than 1/10 of critics declare compensative behaviours, the share of

**Table 4. Compulsive and compensative buyers by attitudes towards online shopping.**

| Segment | (n) | Compulsive buyers | Compensative buyers | Other buying styles | Total |
|---|---|---|---|---|---|
| General population | (1,000) | 3.4% | 12.4% | 84.2% | 100.0% |
| Critics | (530) | 2.6% | 7.6% | 89.8% | 100.0% |
| Sceptics | (148) | 4.1% | 9.5% | 86.4% | 100.0% |
| Potential fans | (100) | 2.1% | 18.1% | 79.8% | 100.0% |
| Fans of online shopping | (128) | 5.3% | 23.2% | 71.5% | 100.0% |

Source: Researcher's own study

compensative buyers among online shopping fans is three times higher and embraces almost 1/4 of the segment.

Again, the one-way ANOVA analysis was conducted to examine if the segments of attitudes towards shopping online differ from each other on the GCBI scale statistically significantly. At the beginning of the analysis, the number of cases within the particular segments were equalled by means of a random selection of the cases. The mean values on the GCBI scale for each randomly formed segment do not differ from the mean values for the whole segments significantly: critics– 28.03; sceptics– 30.21; potential fans– 32.19; fans of online shopping– 37.61. Because the significance of $p < 0.001$ (Levene's test) evidences a dissatisfying statistical significance of the difference between the variance in the tested groups, the Welsch's correction was introduced. The one-way ANOVA indicates at least one pair of mean values differing from each other statistically significantly: $F(3,403) = 19.787$, $p < 0.001$. Hypothesis $H_0$ assuming the same mean values for the examined segments on the GCBI scale can be rejected. The post hoc test shows that the mean values achieved by fans of online shopping on the GCBI scale differ from representatives of all other segments in a statistically significant way (fans of online shopping vs potential fans: mean difference = 5.41, $p < .001$; fans of online shopping vs sceptics: mean difference = 7.40, $p < 0.001$; fans of online shopping vs critics: mean difference = 9.58, $p < 0.001$). At the same time, the mean value achieved by the potential fans on the GCBI scale differs from the analogous ones characterising the critics (potential fans vs critics: mean difference = 4.17, $p < 0.001$). The mean differences between potential fans and sceptics as well as between sceptics and critics are not statistically significant ($p > 0.05$). The above presented data evidence clearly that positive attitudes towards shopping online and susceptibility to compulsive buying co-occur to a greater or lesser extent.

**Frequency of online shopping, attitudes towards online shopping, age, gender, and income as predictors of compulsive buying.** The final analysis aims to answer the research question to what extent the frequency of shopping online, the attitudes towards shopping online, and the basic sociodemographic variables such as age, gender, household income explain susceptibility to compulsive buying. The relations between compulsive buying and the independent variables were examined based on the linear stepwise regression analysis. For this reason, the variable describing expenditures online in comparison with expenditures offline was excluded from the analysis because the variable is strongly correlated with the frequency of online shopping.

Before the analysis, all dependent non-dichotomous variables were recoded to the unique 11-points scales fitting the length of the shortest scale of the attitudes towards online shopping. Next, the predictors were entered in five steps beginning with the variable describing behaviours and attitudes connected with online shopping. The frequency of shopping online was entered first (step 1), then the attitudes towards shopping online (step 2), followed by age (step 3), gender (step 4), and monthly net income of household (step 5). The analysis takes into

**Table 5. Summary of the stepwise regression analysis of frequency of shopping online, attitudes towards shopping online, age, gender, and monthly net income of household as predictors of compulsive buying.**

| Step | B | Std. Error | ß | t | p |
|---|---|---|---|---|---|
| **Step 1** | | | | | |
| Constant | 36.652 | 1.718 | | 21.331 | .000 |
| Frequency of online shopping | .455 | .264 | .097 | 1.723 | .086 |
| **Step 2** | | | | | |
| Constant | 25.680 | 2.116 | | 12.136 | .000 |
| Frequency of online shopping | .809 | .246 | .173 | 3.290 | .001 |
| Attitude towards online shopping | 1.532 | .197 | .409 | 7.761 | .000 |
| **Step 3** | | | | | |
| Constant | 28.312 | 2.780 | | 10.182 | .000 |
| Frequency of online shopping | 0.829 | 0.246 | .178 | 3.372 | .000 |
| Attitude towards online shopping | 1.417 | .213 | .378 | 6.666 | .000 |
| Age | -0.407 | 0.279 | -.082 | -1.455 | .147 |
| **Step 4** | | | | | |
| Constant | 20.446 | 3.072 | | 6.656 | .000 |
| Frequency of online shopping | .783 | .236 | .168 | 3.314 | .000 |
| Attitude towards online shopping | 1.506 | .205 | .402 | 7.352 | .000 |
| Age | -.308 | .269 | -.062 | -1.144 | .253 |
| Gender | 4.455 | .860 | .259 | 5.182 | .000 |
| **Step 5** | | | | | |
| Constant | 23.984 | 3.945 | | 6.080 | .000 |
| Frequency of online shopping | .740 | .238 | .158 | 3.108 | .002 |
| Attitude towards online shopping | 1.451 | .208 | .387 | 6.974 | .000 |
| Age | -.313 | .269 | -.063 | -1.165 | .245 |
| Gender | 4.208 | .876 | .244 | 4.806 | .000 |
| Monthly net income of household | -.359 | .252 | -.074 | -1.426 | .155 |

Step 1: $\Delta R^2 = 0.009$, $p < 0.001$; Std. Error = 8.547; Step 2: $\Delta R^2 = 0.171$, $p < 0.001$; Std. Error = 7.832; Step 3: $\Delta R^2 = 0.177$, $p < 0.001$; Std. Error = 7.817; Step 4: $\Delta R^2 = 0.243$, $p < 0.001$; Std. Error = 7.509; Step 5: $\Delta R^2 = 0.248$, $p < 0.001$; Std. Error = 7.496

Source: Researcher's own study

consideration only online shoppers, meaning those who bought something online during the last 12 months (33% of the sample).

Table 5 presents coefficients after each independent variable was added. It turns out that frequency of shopping online is not the key variable explaining compulsive buying to the greatest extent. The bivariate correlation between the frequency of shopping online and compulsive buying is not statistically significant. The frequency of shopping online begins to play some role if the variable is linked with the attitudes towards shopping online. The more positive attitudes towards shopping online and the more frequent shopping online at the same time, the stronger susceptibility to compulsive buying. The positive attitudes towards online shopping is a condition of the correlation between shopping online and compulsive buying. In other words, exclusively a high frequency of shopping online is not a sufficient factor of compulsive buying, it has to be close-coupled with positive attitudes towards online shopping. Both variables explain 17.1% of the variance of compulsive buying; however, the greatest increase of the R-squared coefficient takes place on step 2 due to the addition of the variable describing attitudes towards shopping online into the model.

Both variables, the frequency of shopping online and the attitudes towards shopping online remain relatively constant and unaffected by the addition of the sociodemographic variables.

Basing on step 2, a growth of susceptibility to compulsive buying by .801 point (+/- .246) on the GCBI scale is expected with each increase of the frequency of shopping online by 1 point. A decidedly stronger effect is observed in case of the attitudes towards shopping online. An increase on the scale of attitudes towards shopping online by 1 point means a growth of 1.532 points on the GCBI. If age is added on step 3, a growth of susceptibility to compulsive buying along with the increasing frequency of shopping by 1 point can be assumed on a nearly the same level as in the previous model (by .829; +/- .246). The results concerning the attitudes towards shopping online keep stable, too—an increase on the scale of the attitudes towards shopping online by 1 point means a growth of 1.417 points on the GCBI scale. Meanwhile, the effect of age on the GCBI scale is statistically insignificant independently of the particular steps. On the one hand, this result deviates from the tendencies evidenced by other empirical studies conducted in Poland [23] and other countries [32–35]. On the other hand, the result is not surprising taking into consideration the fact that users of the e-commerce in Poland are younger by 17 years on average in comparison with non-users of the retail channel (35 y.o. vs 53 y.o.).

If gender is included in the model, the effect of the frequency of shopping online and the attitudes towards shopping online on the GCBI is similar to the previous models. A growth of susceptibility to compulsive buying by .783 (+/- .236) on the GCBI scale can be assumed along with an increase on the scale of shopping frequency by 1 point, while an increase on the scale of the attitudes towards online shopping by 1 point means a growth of 1.506 points on the GCBI scale. The role of gender in the explanation of the GCBI is clear–women show a stronger susceptibility to compulsive buying than men. If a person is a woman, the result on the GCBI scale should increase by 4.455 points (+/- 0.860). Gender, similarly to the scale of attitudes towards shopping online, provides a considerable growth of the explained variance adding a further 6.6% to the explained variance in the prediction of compulsive buying (24.3%).

Adding the monthly net income of household into the model, the above presented effects of predictors on the GCBI scale keep stable. The monthly net income of household does not explain compulsive buying in a statistically significant way at the same time. Again, this outcome might be a result of the relatively consistent sample in respect of age and income, which becomes more differentiated along with age.

To sum up, the meaning of the sociodemographic variables such as age and monthly net income of household for the prediction of compulsive buying is limited. Whether susceptibility to compulsive buying is stronger or weaker depends on the attitudes towards online shopping and gender to a much greater extent. The more positive attitudes towards e-commerce connected with frequent shopping online, the stronger susceptibility to compulsive buying. Without doubts, this effect is stronger among women than among men. Seeking people who buy compulsively to the greatest extent, women buying online very frequently and showing very positive attitudes towards e-commerce should be taken into consideration first of all.

## Discussion

In accordance with our expectations, we observe the phenomenon of compensative and compulsive buying in Poland although the prevalence of this type of consumer behaviour is weaker than in West Europe or in comparison with the USA. The discussion about the reasons for the less intensive development of compensative and compulsive buying was held elsewhere [23]. Independently of the weaker prevalence, the identified co-occurrence of online shopping and susceptibility to compulsive buying is coherent with the results coming from other studies. There exist at least a few plausible explanations why the correlation between online shopping and compulsive buying appears.

Firstly, compulsive buying is a type of a behavioural addiction meaning "an irresistible desire of a specific state of experience" [39, p. 20]. In case of compulsive buying, the subject of the addiction can be defined as a purchase act bringing forth relief of tension which arises on account of negative experiences of everyday life. A purchase act of a compulsive buyer serves escaping from the reality. From the sociological point of view, the subject of an addiction is not a material good (such as a substance or a consumer good) but the interaction between an addicted person and their addiction subject which appears in specific circumstances [40]. Hence, an easy access to the subject of addiction is a crucial condition for the development of the addiction. Because only a purchase act is the addiction subject of compulsive buying, shopping online easily accessible day and night renders more favourable conditions for the development of compulsive buying than shopping offline.

Secondly, the development of consumer society supports indirectly the positive correlation between shopping online and compulsive buying. Members of consumer society believe that "meaning and satisfaction in life are to be found through the purchase and use of consumer goods" [41, p. 4]. For this reason, some traits of consumer society are developed to a greater extent than in case of traditional society. Individualization coupled with self-actualization is one of the processes which intensively go off in consumer society especially [42]. Consumeristic struggle to distinguish oneself leads to a far-reaching differentiation of lifestyles basing among others on consumer symbolic values contained in material goods, services, and places of shopping. Representatives of at least two types of life styles might be especially inclined to an excessive use of shopping online–consumers strongly oriented on fashion and so-called smart shoppers. Online shopping offers unending possibilities for both target groups to achieve their consumer goals. Thanks to online shopping persons representing very positive attitudes towards fashion have a chance to find original or even inimitable pieces of clothes, footwear or accessories in an easier and faster way than in stationary retail. Smart shoppers seeking bargains more willingly use online than stationary retail, because the Internet search engines provide a swift comparison of offers, promotions, sales etc. A smart purchase can be realised quickly and effortlessly.

Thirdly, social control is an important instrument braking the development of substance and behavioural addictions. Social control is a set of mechanisms preserving conformity between human behaviours and social norms rooted in social values [43]. This mechanism consists in direct and indirect sanctions depending on the type of norms. Direct sanctions are used by government or in a broader sense by state if legal norms are violated. Indirect sanctions understood as pressure of the public opinion are used by social groups to deter the individuals who might violate moral, custom, or fashion norms potentially. Certainly, hiding a violation of the consumer norm of rational buying from e.g. relatives is easier in the virtual than in the real world. Shopping online might be conducted in the meantime e.g. doing the home office. Then, these activities do not arouse suspicions among relatives; the development of compulsive buying might remain hidden.

The results of the data analysis showed that the positive correlation between shopping online and compulsive buying is stronger among women than among men. The result is coherent with most studies which point to a stronger susceptibility of women to compulsive buying than of men [22, 27, 32–34]. Schernhorn, Reisch, and Raab [21] explained the phenomenon by the socialization process, which supports more passive and emotional ways to manage stress and conflicts among women. They are characterised by inclinations to solve problems without publicity and in socially accepted ways. Shopping is one of the ways which is not only accepted in consumer society but it is even desirable as a source of social prestige. In addition, in traditionally oriented societies the specific socialization of a woman's role causes that women are prepared to look after household to a bigger extent than men. Hence, when

women purchase and derive pleasure from shopping, it is socially more accepted than in case of men. For this reason, susceptibility to compulsive buying among women might be greater than among men and compulsive buying among women can be practised unnoticeably even until financial troubles of household appear.

## Conclusions

The presented study conducted in 2019 on the sample of 1,000 Poles aged 15 years and over pointed out the shares of compulsive and compensative buyers in the general population on the level near to 4% and 16%. According to the bivariate data analysis, susceptibility to compulsive buying is stronger among online than offline shoppers. In addition, susceptibility to compulsive buying increases along with the growing frequency of shopping online, growing expenditures on online shopping, and more positive attitudes towards online shopping. The regression analysis proves that high frequency of shopping online goes with susceptibility to compulsive buying under the condition of positive attitudes towards shopping online. This effect is stronger among women than among men.

The real range of both types of consumer behaviours may be significantly greater, because the measured ratios describe only those consumers who are conscious of their compensative and compulsive behaviours. Meanwhile, some buyers may be unaware that they buy compulsively or compensatively or they deny the fact. This conclusion concerns all shoppers independently of the fact if they do the shopping in online or stationary retail. For the same reason, the real correlation between shopping online and compulsive buying might be stronger.

Although the obtained data do not allow the causative conclusions, it is worth considering the possible directions of the causality. An educe that online shopping might cause compulsive buying is rather false. It seems that online shopping is only a circumstance which strengthens the influence of real factors of compulsive buying such as low self-esteem or strong materialism orientation. Only the fact of the purchase acts conducted in an online marketplace cannot cause susceptibility to compulsive buying. In contrast, the reverse direction of the causality seems to be rather possible. Indeed, compulsive buyers might do the online shopping more willingly compared with other types of buyers for several reasons. Firstly, the online marketplace offers an easier development of susceptibility to compulsive buying among compulsive buyers than stationary retail. Compulsive buying consists in addiction to compensation delivered by purchase acts. It is natural that compulsive buyers choose online shopping willingly because of the unlimited access to the global marketplace. Secondly, compulsive buyers who do the shopping online can keep the truth about their addiction from relatives for an extended period. It is worth underlying a possible effect of a reverse coupling if compulsive buyers do the online shopping. On the one hand, compulsive buying might be an important factor of an extensive online shopping, on the other hand, the extensive online shopping done by compulsive buyers might strengthen their addiction.

Currently, we urgently need more studies on this issue. The COVID-19 pandemic creates conditions contributing to the development of compensative and compulsive buying in a double sense. Firstly, the lockdowns consisting in the limitation of stationary retail force more intensive usage of online shopping. Because the hypothesis is confirmed that online shopping supports the development of compulsive buying to a greater extent than stationary retail, an increasing share of compulsive buyers in the general population should be observed in the future. Secondly, social isolation due to the COVID-19 pandemic causes real negative psychical consequences among people connected with the state of the chronic stress such as anxiety state or mood disorders inducing a decreasing level of self-esteem. Initially, compensative and subsequently compulsive online buying conducted easily under the conditions of the social

isolation might be treated by consumers as an escape from the problems of everyday life during the COVID-19 pandemic.

The obtained data allow to draw some conclusions for consumer policy. The results show, especially among women, that high frequency of shopping online goes with susceptibility to compulsive buying under the condition of positive attitudes towards this kind of shopping. The positive correlation between online shopping and susceptibility to compulsive buying, increase of the e-commerce market as a result of lockdowns limiting stationary retail, and consequences of the COVID-19 pandemic, the latter at the same time being primary factors of compulsive buying such as weakened self-esteem arising from social isolation, allow anticipating a considerable growth of online compulsive buying in the future. It seems to be necessary to launch educational programmes about reasons and negative consequences of compulsive buying especially dedicated to younger consumers who are subjected to the addiction to a greater extent than older ones. The e-commerce enterprises, first of all the big brands, could play a special role in this area. Thanks to the educational measurements framing responsible consumption, the companies could strengthen the protection of their customers against negative consequences of excessive buying, which finally would support the Public Relations of the brands and its owners. Without doubts, the development of the special education programmes would be in line with the main tasks of consumer policy–education, information, and protection [44].

## Supporting information

**S1 File. Questionnaire PL.**
(PDF)

**S2 File. Questionnaire ENG.**
(PDF)

## Author Contributions

**Conceptualization:** Grzegorz Adamczyk.

**Data curation:** Grzegorz Adamczyk.

**Formal analysis:** Grzegorz Adamczyk.

**Funding acquisition:** Grzegorz Adamczyk.

**Investigation:** Grzegorz Adamczyk.

**Methodology:** Grzegorz Adamczyk.

**Project administration:** Grzegorz Adamczyk.

**Resources:** Grzegorz Adamczyk.

**Software:** Grzegorz Adamczyk.

**Supervision:** Grzegorz Adamczyk.

**Validation:** Grzegorz Adamczyk.

**Visualization:** Grzegorz Adamczyk.

**Writing – original draft:** Grzegorz Adamczyk.

**Writing – review & editing:** Grzegorz Adamczyk.

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
