## [Decision Letter · Decision Letter 0]

25 Mar 2021

PONE-D-21-01942

Compulsive and Compensative Buying Among Online Shoppers. An Empirical Study

PLOS ONE

Dear Dr. Adamczyk,

Thank you for submitting your manuscript to PLOS ONE. After careful consideration, we feel that it has merit but does not fully meet PLOS ONE’s publication criteria as it currently stands. Therefore, we invite you to submit a revised version of the manuscript that addresses the points raised during the review process.

We look forward to receiving your revised manuscript.

Kind regards,

Frantisek Sudzina

Academic Editor

PLOS ONE

Journal Requirements:

4. We note you have included a table to which you do not refer in the text of your manuscript. Please ensure that you refer to Tables 1, 2, 3 in your text; if accepted, production will need this reference to link the reader to the Table.

5.  Thank you for stating the following in the Financial Disclosure section:

"This work was supported by the Institute of the Public Opinion and Market Research GfK Polonia Sp. z o.o. belonging to the GfK Group. Thanks to the support I added the GCBI scale to the GfK's regular study about shopping online in Poland "

We note that you received funding from a commercial source: GfK Polonia Sp. z o.o.

Reviewers' comments:

Reviewer's Responses to Questions

**Comments to the Author**

1. Is the manuscript technically sound, and do the data support the conclusions?

Reviewer #1: Yes

Reviewer #2: Yes

2. Has the statistical analysis been performed appropriately and rigorously? 

Reviewer #1: Yes

Reviewer #2: Yes

3. Have the authors made all data underlying the findings in their manuscript fully available?

Reviewer #1: Yes

Reviewer #2: Yes

4. Is the manuscript presented in an intelligible fashion and written in standard English?

Reviewer #1: Yes

Reviewer #2: Yes

5. Review Comments to the Author

Reviewer #1: Overall the study is fine. I feel it might be making a bit of a 'causal' claim that online buying leads to compulsive behavior? We don't know if some people who have some issue with compulsive behaviour are the ones that do a lot of online shopping, or maybe the act of online shopping builds up some dependency or need for gratification over time -> compulsive. Perhaps you could canvass the idea that effects could be going in both directions.

Some minor english expression issues - you could, if you wanted, get an english language copyeditor because there are a lot of minor inconsistencies with the english. A specific suggestion is:

Currently:

Poles aged 15 years old and more splitting into users and non-users of the e-commerce market offer. Secondly, the conducted analysis shows to what extent the prevalence is differentiated by the frequency of online shopping, by the extent of the expenditures on online shopping compared with offline shopping, by attitudes towards online shopping, and by sociodemographic conditions (gender, age, monthly net income of household).

Re-write as:

Poles aged 15 years old and over, with the sample split into users and non-users of the e-commerce market offer. Secondly, the conducted analysis shows to what extent the prevalence of compulsive and compensative buying is differentiated by the frequency of online shopping, by the extent of expenditures on online shopping compared with offline shopping, by attitudes towards online shopping, and by sociodemographic conditions (gender, age, monthly net income of household).

Page 1 line 38 not differentiations, differences would be a more appropriate word

p. 2 Line 42, shopping and doing the online shopping very frequently. – delete the word ‘the’ here

Reviewer #2: To the authors;

The idea of examining "the phenomenon of compensatory and compulsive buying among online shoppers" is particularly suggestive and of undeniable contemporary interest. The sample of the study is also another important strength. Overall, in our opinion, this is a good paper. We suggest some minor changes that we hope will contribute to improve this work.

Introduction

- Although the authors refer to the relevance of online shopping in studies carried out in different countries, the importance of this phenomenon at a global level requires the inclusion of some other recent bibliographic references from studies carried out in Asian countries (where its high prevalence is reported). Specifically, some bibliographic reference could be added in lines 72-74, where it is said that: "shopping online is an important universal factor of compulsive buying independent of cultural conditions".

- Specify to which scale the authors refer in line 132.

- Justify the extensive exposition in the introduction of the variables self-esteem and materialism as factors "strengthening susceptibility to compulsive buying", when the authors do not include them as variables to be analyzed in this research (lines 135-167).

Material and method

- It would be convenient to detail more the characteristics of the sample (sex, age,...). It is only indicated that "the study was carried out on the statistically representative sample of 1,000 Poles aged 15 years old and over" (lines 296-297).

- Is it necessary to refer to the differences between different measures of compulsive buying in the paragraph that presents the self-report used for the assessment of compulsive buying (see lines 303-309)

- It would be advisable to indicate in advance which statistical analyses are going to be applied before presenting the results.

Results

- Include the reference to the various tables of results (tables 1, 2 and 3 -lines 391, 442, 513).

- We suggest to include the description of the statements that evaluate the attitudes towards shopping online in the material section, not in the results section (line 444), as it has been done with the description of the German Compulsive Buying Indicator (line 303).

Conclusions

- It would be interesting if the authors, based on the findings, could suggest some specific guidelines for action and/or what implications may have for the design of future prevention/intervention programs for compulsive online shopping.

References

- We recommend to review the inclusion of references in the text, since on many occasions the bibliographic references are duplicated.

6. PLOS authors have the option to publish the peer review history of their article (what does this mean?). If published, this will include your full peer review and any attached files.

Reviewer #1: **Yes: **John Dawes

Reviewer #2: No

---

## [Author Response · Author response to Decision Letter 0]

16 May 2021

The file with answers is attached

---

## [Editor Report · Decision Letter 1]

19 May 2021

Compulsive and compensative buying among online shoppers. An empirical study

PONE-D-21-01942R1

Dear Dr. Adamczyk,

We’re pleased to inform you that your manuscript has been judged scientifically suitable for publication and will be formally accepted for publication once it meets all outstanding technical requirements.

Kind regards,

Frantisek Sudzina

Academic Editor

PLOS ONE
---

## [Editor Report · Acceptance letter]

24 May 2021

PONE-D-21-01942R1 

Compulsive and compensative buying among online shoppers. An empirical study 

Dear Dr. Adamczyk:

I'm pleased to inform you that your manuscript has been deemed suitable for publication in PLOS ONE. Congratulations! Your manuscript is now with our production department. 

Kind regards, 

on behalf of

Dr. Frantisek Sudzina 

Academic Editor

PLOS ONE